# Dietary Insect Powder Protein Sources Improve Protein Utilization by Regulation on Intestinal Amino Acid-Chemosensing System

**DOI:** 10.3390/ani10091590

**Published:** 2020-09-07

**Authors:** Hongnan Liu, Bie Tan, Xiangfeng Kong, Jianjun Li, Guangran Li, Liuqin He, Miaomiao Bai, Yulong Yin

**Affiliations:** Hunan Provincial Key Laboratory of Animal Nutritional Physiology and Metabolic Process, National Engineering Laboratory for Pollution Control and Waste Utilization in Livestock and Poultry Production, Key Laboratory of Agro-Ecological Processes in Subtropical Region, Hunan Provincial Engineering Research Center for Healthy Livestock and Poultry Production, Scientific Observing and Experimental Station of Animal Nutrition and Feed Science in South-Central, Ministry of Agriculture, Institute of Subtropical Agriculture, Chinese Academy of Sciences, 644 Yuanda 2 Road, Changsha 410125, Hunan, China; liuhn@isa.ac.cn (H.L.); jianjunli@isa.ac.cn (J.L.); womumu1990@163.com (G.L.); heliuqin@hunnu.edu.cn (L.H.); miaomiaobai1115@126.com (M.B.); yinyulong@isa.ac.cn (Y.Y.)

**Keywords:** insects, protein resource, amino acid transport, piglets

## Abstract

**Simple Summary:**

Insect powders, including *Tenebrio molitor* (TM), *Musca domestica* larvae (MDL) and *Zophobas morio* (ZM), as high-quality and renewable protein sources are commonly applied in livestock and poultry feed production. The molecular effect of insect protein on amino acid metabolism in pigs needs to be explored. We found that insect powder as a protein source in feed regulated the mTOR signal pathway and improved amino acid transportation in the intestine for growth promotion. Insect powder may be a potentially promising protein source for pig production.

**Abstract:**

This study was conducted to evaluate the effects of dietary insect powder supplementation as a protein source on plasma amino acid profiles, intestinal amino acid transport and sensing in a piglet model. A total of 144 weanling piglets were randomly assigned to four experimental diets for two phases (Days 1–28 and Days 29–56), to assess the effects on amino acid profiles and transportation in the segments of the intestine. The groups were basal diet (control), control diet plus *Tenebrio molitor* (TM), control diet plus *Musca domestica* larvae (MDL) and control diet plus *Zophobas morio* (ZM). The plasma free amino acid levels were stable comparable among treatments, except that the lysine level was significantly reduced by dietary MDL and ZM supplementation in the first phase (*p* < 0.05). In the 1st phase, the sensitivity of intestinal segments to the regulation of the amino acid level by insect powder supplementation follows sequence: colon > ileum > jejunum, while the order switched to jejunum > colon > ileum in the 2nd phase. The relative RNA expressions of mitogen-activated protein 4 kinase 3 (MAP4K3), sodium dependent neutral amino acid transporter2 (SNAT2), the transient receptor potential cation channel subfamily V member 1 (TRPV1) and taste 1 receptor member 1/3 (T1R3) in the segments of the intestine were affected by different dietary insect powder supplementation. G protein-coupled receptor family C group 6 member A (GPRC6A) level in the jejunal and colonic mucosa was upregulated by MDL supplementation (*p* < 0.05). These results indicated that dietary insects improved the metabolism of the amino acid in the prophase (the 1st phase) through regulating the sensing gene and mTOR signal pathway in intestinal mucosa by targeting different receptors. The finding demonstrates that the insect powder is a potentially promising source for protein deposition.

## 1. Introduction

Protein deposition relies on the use of a high-quality protein diet. A high-quality protein source must have adequate amino acid distribution and high digestibility, as well as high content protein [1]. Insects can turn low-grade biowaste into high quality proteins indicating that insects as a renewable protein rich feed ingredient is technically feasible. [2]. At present, about one million species of insects are known in the world, whereas the total species of insects may reach 80 million including the unknown ones [3]. Previous studies showed that most insects had high nutritional qualities as human food [4,5]. Insect protein appears as a potentially suitable ingredient with several characteristics that make it relevant for its use in the diet, such as its high protein level [6], good amino acid profile [7] and its characteristics in terms of secure supply with less environmental impact [8]. Moreover, insects can be reared on low-grade biowaste and can turn biowaste into high-quality proteins. However, the molecular effect of dietary insects on amino acid transportation is not well known.

*Tenebrio molitor* (TM), *Musca domestica* larvae (MDL) and *Zophobas morio* (ZM) belong among the most common species of insects as feedstuff in the animal food market, and they are characterized by rapid reproduction, large biomass, less investment of production and seasonal reproduction for large-scale production. These insects contain a good amino acid profile, found in the previous study [9]. Amino acids (AA) are well known to represent the units used for protein biosynthesis and are also precursors for the synthesis of functional molecules, such as peptides, hormones, neurotransmitters, purine, pyrimidine nucleotides, etc. [10,11]. Plasma AA level approximately reflects the AA composition in the form of protein in the diet [12]. Tissue AA level shows the pros and cons of the dietary AA pattern [13]. The variation of the amino acid level leads to the corresponding cellular response. By the relevant signaling system-mediation and regulation on the downstream induction factor, protein metabolism changes dynamically, resulting in the improvement of production performance macroscopically [14]. In the previous study, we found that plasma concentrations of total protein and albumin were reduced by dietary ZM whereas methionine (Met) apparent ileal digestibility (AID) was improved in the pigs [9]. Based on that context, the present study evaluated the potential value of different insect powder as a feed ingredient used for early-weaned piglets. However, the amino acid transportation in the intestine, and the potential value of different insect powder are still unclear. In this study, we hypothesize that insect powders supplementation regulates the free amino acid profiles in plasma by affecting the amino acid transporter and sensing gene expression in the intestinal mucosa of the pig model.

## 2. Materials and Methods

The experimental procedures for this study were approved by the Animal Care and Use Committee of the Institute of Subtropical Agriculture, Chinese Academy of Sciences (2016-8A).

### 2.1. Animal Experiment Design

The animal experimental design was conducted according to the description in the previous study [9]. This study was based on the same experimental protocol and piglets. As in Ji et al., one hundred and forty-four piglets (Duroc × Landrace × Yorkshire, 4.74 ± 0.01 kg body weight) were randomly assigned to four groups. Each group had 36 piglets arranged in 6 replicates of 6 piglets. They were supplemented with a soybean-based diet +5% of plasma protein powder (control), a control diet +5% of the TM powder, a control diet +5% of the MDL powder or a control diet +5% of the ZM powder, respectively. The experiment was divided into two phases. In both of the first (Days 1–28) and second phase (Days 29–56), levels of the crude protein in the diets were 21% and 17%, respectively.

### 2.2. Sample Collection and Measurement

On Days 28 and 56 after initiation of dietary supplementation with insect powder, one piglet per replicate was chosen randomly (half of the barrows and half of the gilts in each treatment), and 5 mL blood samples from the precaval vein were withdrawn into heparinized tubes. Blood was centrifuged for 15 min at 3000× *g* and 4 °C. Plasma was collected and stored at −20 °C. The sampled piglets were euthanized for collecting the sample of jejunum, ileum and colon of the intestine for 2 cm. Mucosa was collected carefully from the tissue using cover glass. Samples were quick-frozen in liquid nitrogen and stored at −80 °C. The level of AA in plasma, jejunal mucosa, ileal mucosa, and colonic mucosa were calculated according to the method described by Yin et al. (2000) [15].

### 2.3. Real-Time PCR

Total RNA was isolated from jejunum, ileum and colon samples using a Trizol Reagent (Invitrogen, Carlsbad, CA, USA) following the manufacturer’s instructions. The extracted RNA was dissolved in diethylpyrocarbonate (DEPC)-treated water, and its concentration was assessed using an Eppendorf Biophotometer (Eppendorf AG, Hamburg, Germany) and its integrity verified by electrophoresis on a 1% agarose gel. After DNase I treatment (Takara, Otsu, Japan), 1 μg of total RNA was used as a template for cDNA synthesis using an Oligo (dT) primer (Takara, Japan). The resultant cDNA was diluted and used for evaluating gene expression.

All primers were developed previously for the amplification of mRNA sequences of pig (Sussucrofa, Appendix A). Quantitative real-time PCR (qPCR) for the target genes TRPV1 (The transient receptor potential cation channel subfamily V member 1), FFAR3 (free fatty acid receptor 3), TRPM5 (transient receptor potential cation channel subfamily M member 5), mGluR (metabotropic glutamate receptors), T1R3 (taste 1 receptor member 1/3), PLCβ2 (Gβγ-mediated phospholipase C β2), GPRC6A (G protein-coupled receptor family C group 6 member A), PAT1 (proton-coupled amino acid transporter 1), PAT2, Y+LAT1 (L-type amino acid transporter 1), SNAT2 (sodium dependent neutral amino acid transporter 2), GCN2 (general control nonderepressible 2), MAP4K3 (mitogen-activated protein 4 kinase 3), PIK3C3 (phosphatidylinositol 3-kinase catalytic subunit type 3), S6K1 (ribosome protein subunit 6 kinase 1) and the housekeeping gene (β-actin) were performed in a 10 μL reaction volume including 1 μM of each forward and reverse primer, 2 μL of cDNA, 2 μL of DEPC-treated water and 5 μL of SYBR Premix Ex Taq (Takara Bio Inc., Tokyo, Japan). The qPCR was carried out (Lightcycler-480I I, Roche Diagnostics GmbH, Mannheim, Germany) with the following conditions: 95 °C for 30 s, 40 cycles of denaturation at 95 °C for 15 s, annealing at 60 °C for 30 s and elongation at 72 °C for 30 s, followed by a melting curve analysis. The relative expression of target genes was expressed as 2^−ΔΔCt^, where ΔΔCt = (CtTarget – Ctβ-actin) treatment – (CtTarget – Ctβ-actin) control.

### 2.4. Statistical Analysis

Data were expressed as the means ± SEM. Results were statistically analyzed using a one-way ANOVA (SPSS18.0, SPSS Inc., Chicago, IL, USA). The Duncan’s multiple range test was used to compare differences among the treatment groups. A *p*-value of less than 0.05 was taken to indicate statistical significance.

## 3. Results

### 3.1. Free Amino Acids Profiles in Plasma

As shown in Table 1, plasma Val, Tyr and Lys levels were significantly affected by dietary insect supplementation at the end of the 1st phase of the experiment (*p* < 0.05). The plasma Val level in pigs fed the TM diet was higher than that fed the MDL and ZM diets (*p* < 0.05). Compared to the control diet, MDL and ZM had higher Lys in plasma (*p* < 0.05). In the 2nd phase, the result showed that dietary insect powder supplementation changed the level of Cys, Ile and Lys in the plasma, whereas Cys content in the ZM group was increased significantly compared to the control group (*p* < 0.05).

### 3.2. Free Amino Acids Profiles in the Intestinal Mucosa

In the jejunal mucosa, Cys and Lys contents were changed by the diet treatment during the 28-day experiment (*p* < 0.05). Compared to the control group, all of the insect powder supplementations decreased the Cys level, whereas MDL reduced the Lys level (*p* < 0.05). In the 2nd phase, the results showed that the contents of fifteen amino acids were reduced by the supplemented insect powder, excluding Glu and Cys (*p* < 0.05). Compared to control group, the reduction of the amino acids regulated by the diet treatment was in line with the trend: MDL > TM > ZM (Table 2).

Amino acids profile in the ileal mucosa was shown in Table 3. The level of amino acids was affected by insect powder supplementation in the 1st phase (*p* < 0.05), excluding Asp, Glu, Gly and Ala, whereas Thr, Glu and Gly were the only AAs that were unaffected by the dietary insect powder in the 2nd phase. Based on the results of both phases, we found that MDL and ZM significantly increased the level of amino acids in ileal mucosa compared to the control group (*p* < 0.05), while no significant change was observed in the TM group.

In the colonic mucosa, the level of amino acids was changed (*p* < 0.05) by the dietary insect powder supplementation on the 28th day, excluding Thr and Gly. As the result in jejunal mucosa, the capability of MDL to upregulate amino acid levels was better than that of the other two groups with insect supplementation. In the 2nd phase, the levels of Ser, Met, Met, Tyr, Phe, Lys, Arg and Pro were affected by the dietary insect powder (*p* < 0.05). Unlike the result in the 1st phase, the three insect powders were comparable to the deposition of these amino acids in the 2nd phase (Table 4).

### 3.3. Amino Acid Transporter and Sensing Gene Expression in Intestinal Mucosa

In the jejunal mucosa, the relative RNA expressions of GPRC6A and MAP4K3 were affected by dietary insect powder supplementation at the 1st phase (*p* < 0.05). Compared to the control group, the RNA level of GPRC6A in ZM was upregulated whereas MAP4K3 in TM group was reduced (*p* < 0.05). In the 2nd, mGluR, T1R3, PLCβ2, PAT2 and S6K1 expressions were regulated, as well as GPRC6A and MAP4K3 (*p* < 0.05). Compared to the control diet, dietary TM upregulated S6K1 gene expression, and dietary MDL downregulated GPRC6A, whereas ZM reduced the expression of PLCβ2 and GPRC6A in the mucosa (*p* < 0.05; Table 5).

In the ileal mucosa, no significant change was observed on the expression of amino acid transporters and the sensing gene at the 1st phase. In the 2nd phase, meal treatment changed the gene expression of TRPV1, T1R3 and PLCβ2 (*p* < 0.05). Compared to the control diet, supplemented MDL upregulated TRPV1, T1R3 and PLCβ2 expression (*p* < 0.05), but not TM or ZM (Table 6).

Similar to the results in the ileal mucosa, no significant change was observed in the colonic mucosa at the 1st phase (Table 7). However, GPRC6A and SNAT2 expression were regulated by dietary insect supplementation (*p* < 0.05). Compared to the control diet, supplemented TM or MDL reduced the GPRC6A expression, whereas ZM upregulated the SNAT2 expression (*p* < 0.05).

## 4. Discussion

Insect is a source of protein, of which a high quantity (30–70%) is contained in the dry material. The protein from the insect is high in essential amino acids, which are deemed to have favorable conversion efficacy [16]. In the previous study, we found that the AID of Met was increased by dietary insect powder in piglets. Met is the second limiting AAs for swine, which attend in protein synthesis and sulfur metabolism. In our current study, we hypothesized that the addition of insect powder may improve the amino acid profile by regulating their transportation in the intestine using the pig model. We sought to evaluate the effects of insect powder on the amino acids transporter and sensing gene expression of swine. Base on the results, a reduction of the Lys level in the plasma were found in pigs fed the ZM powder in the whole experiment, whereas the reducing Lys level was also observed in pigs fed the MDL powder in the 28-day phase. Lysine is the first limiting AA in pig diets based on cereal-soybean meal ingredients [17]. Free lysine is known to be used efficiently for growth and protein deposition [18]. In the profile of plasma and various intestinal segments, Val and Tyr were the other two AAs regulated by dietary insects.

To further determine amino acid transportation in the intestine, the amino acid transporter and sensing gene expression was tested. The gastrointestinal tract of animals is capable of sensing and recognizing nutrients, as well as initiating digestive, absorption and metabolic cascades. Most nutrient receptors are distributed on enteroendocrine cells. These important nutrient receptors include membrane-bound solute carriers (SLCs), G-protein-coupled receptors (GPCRs) and intracellular receptors. Amino acid transporters are cell surface receptors that directly trigger nutritional signals in response to amino acid levels. Converted into a chemical signal, amino acids cause signal transduction by changing the binding transporter or the conformation with the transport protein [19]. In the segment of intestine, the activity of protein trends to vary with growth, especially for chemosensors and transporters [20,21,22]. In this study, we examined the expression of amino acid transporters PAT1, PAT2 and y+LAT1 in the GI tract. PAT2 and its paralog, PAT1/LYAAT-1, are transporters for small amino acids such as Gly, Ala and Pro [23], whereas y+LAT1 is for alkaline and neutral amino acids such as Leu, Arg, Lys, Gln and His.

Amino acids transporters, such as PATs, LATs and SNATs, have different characteristics on the transport substrate, drivers and affinity, and are responsible for different varieties of amino acids in intestinal tissues. As a member of the phosphatidylinositol 3-kinase-related kinase family, the mammalian target of rapamycin (mTOR) integrates the input from amino acids in various tissues and regulates cell growth and protein synthesis in mTOR-S6K-4W-BP1. Proton-assisted amino acid transporter (PAT), a member of the SLC36 family, transports small amino acids (glycine, alanine and proline) [24]. Sodium-coupled neutral amino acid transporter 2 (SNAT2) shares substrates (Ala and Pro) with PAT2. Encoded by the SLC38A2 gene, mRNA and protein expression of SNAT2 was elevated by the increase of essential amino acid. It was dependent on the mTOR pathway and may be an adaptive mechanism for the increasing pressure of intracellular amino acid transportation [25]. Besides, the substrate species also affects the expression of SNAT2. The inhibitory effect of the substrate on SNAT2 is positively correlated with the substrate and SNAT2 transport Km [26]. mTORC1 and GCN2 control the sensing signaling pathway in which AAs are transported into or out of the membrane, respectively [25]. In the current study, we found that mRNA expression of SNAT2 was activated without change on GCN2 in the colon by supplemented ZM. ZM supplementation provides a change of the amino acid content and variety, as it may regulate the mRNA expression of SNAT2 through the mTOR signal pathway, resulting in AA transportation into the membrane. The determination of signaling molecules, and the mRNA and protein levels associated with the aforementioned nutrient-sensing signaling pathways in the colonic mucosa has revealed that the addition of insect powder to fodder enhanced the expression of genes related to amino acid transport and sensing, as well as the mTOR signaling pathway in colonic mucosa, indicating that the insect powder could facilitate nutrient utilization and protein metabolism.

As a chemical signal, AAs activate intracellular adenylate cyclase (AC) to produce cAMP and protein kinase A (PKA) to close the K^+^ channel. The reaction leads to depolarization of the cell membrane, extracellular Ca^2+^ influx causing an increase in intracellular free Ca^2+^ concentration and triggering the release of neurotransmitters [27]. Besides, extracellular signals can also bind to G-protein coupled receptors (GPCRs), and activate phospholipase C (PLC). It can hydrolyze 4,5-diphosphophosphatidylinositol (PIP2), resulting in the IP3-gated calcium channel opened on the calculus membrane, and the release of Ca^2+^ in turn activates TRPM5 to promote membrane depolarization.

Binding with glutamate, the metabotropic glutamate receptor (mGluR) is a member of GPCRs. *GPRC6a* is a protein that recognizes Arg and Lys in the gastrovascular cavity, with the highest expression in the jejunum and colon [28]. It has been hypothesized that GPRC6a requires calcium ions for amino acid sensing [29]. MAP4K3, belonging to the Ste20-related kinase family, is required for amino acids to activate S6K and induces phosphorylation of the mTOR-regulated inhibitor [1]. It is a highly conserved serine/threonine kinase that participates in interconnections between multiple signaling pathways, including the IMD, EGFR, TORC1 and JNK signaling pathways [7]. Previous studies have shown that mTORC1 can be regulated by amino acid concentrations via MAP4K3 activity [1]. In the current study, the change in gene expression of GPRC6a, MAP4K3, mGluR, PLCβ2 and S6K1 regulated by different insect powder supplementation indicated that the protein sources from an insect might regulate the amino acids file through activating GPCRs rather than SLCs in the jejunum.

The picture was different in ileal mucosa. Even the insect supplementation brought tremendous changes in the amino acids file, TRPV1 and T1R3 were the remaining sensors whose mRNA expressions were upregulated in the MDL treatment. T1R1/T1R3 recognizes aliphatic amino acids and is especially sensitive to Gln and Asp [30]. Phe, Trp and Lys were also found to upregulate the expression of T1R1/T1R3 in mouse STC-1 cells [31]. TRPV1 and T1R1/T1R3 are important amino acid sensing receptors. T1R1/T1R3 is a receptor that directly senses energy levels and amino acid concentrations. Previous studies have indicated that the knockout of genes encoding T1R1/T1R3 could directly affect the amino acid-dependent mTORC signaling process [32]. Glutamate, glucose and some artificial sweeteners are capable of activating T1R1/T1R3, which in turn activates PLCβ2 via Gg to produce DAG and IP3 [33]. IP3 triggers the release of intracellular Ca^2+^ by binding to IP3R3, which in turn induces Na^+^ influx by activating the TRPM5 channel, eventually leading to membrane depolarization and neurotransmitter release [34]. The desensitization of TRPV1 implicated various signaling pathways such as calmodulin and calcineurin, and the decrease of PIP2 [35]. Together, the change on gene expression of TRPV1 and T1R3 indicated that the dietary *Musca domestica* larvae powder might affect AA transportation through the T1R3-TRPV1-PIP2 signal pathway in the ileum.

## 5. Conclusions

Our findings showed that the use of insect powder as a protein source in feed improved amino acid transportation in the intestine through regulating the sensing gene and mTOR signal pathway in intestinal mucosa by targeting different receptors, which indicates that insect powder is a potentially promising protein source benefiting of utilization.

## Figures and Tables

**Table 1 animals-10-01590-t001:** The effect of different insects protein on plasma free amino acids profiles in weanling pigs (nmol mL^−1^).

Items	Treatments ^1^				SEM	*p*-Value
	Control	TM	MDL	ZM		
Day 28						
Asp	0.10	0.11	0.11	0.12	0.01	0.740
Thr	0.56	0.50	0.58	0.42	0.05	0.211
Ser	0.19	0.22	0.19	0.22	0.02	0.548
Glu	0.60	0.76	0.74	0.72	0.07	0.377
Gly	1.46	1.52	1.45	1.58	0.15	0.929
Ala	1.06	1.20	1.12	0.85	0.12	0.251
Cys	0.04	0.05	0.05	0.04	0.01	0.674
Val	0.32 ^a,b^	0.35 ^a^	0.28 ^b^	0.28 ^b^	0.02	0.042
Met	0.08	0.08	0.08	0.07	0.01	0.659
Ile	0.14	0.15	0.15	0.14	0.01	0.743
Leu	0.29	0.28	0.26	0.24	0.02	0.404
Tyr	0.10 ^a,b^	0.06 ^b^	0.14 ^a^	0.14 ^a^	0.02	0.037
Phe	0.10	0.10	0.11	0.10	0.01	0.757
Lys	0.27 ^a^	0.23 ^a,b^	0.19 ^b^	0.18 ^b^	0.02	0.047
His	0.32	0.37	0.33	0.36	0.02	0.854
Arg	0.24	0.23	0.23	0.19	0.02	0.256
Pro	0.31	0.32	0.37	0.32	0.02	0.189
Day 56						
Asp	0.08	0.06	0.07	0.07	0.01	0.565
Thr	0.25	0.29	0.20	0.36	0.05	0.456
Ser	0.15	0.15	0.13	0.16	0.02	0.834
Glu	0.67	0.56	0.57	0.66	0.06	0.256
Gly	1.05	1.13	1.15	1.10	0.12	0.934
Ala	0.71	0.79	0.61	0.82	0.08	0.327
Cys	0.03 ^a^	0.04 ^a,b^	0.05 ^a,b^	0.06 ^b^	0.01	0.040
Val	0.23	0.19	0.24	0.20	0.02	0.536
Met	0.03	0.04	0.03	0.04	0.01	0.643
Ile	0.10 ^a,b^	0.08 ^a,b^	0.11 ^a^	0.08 ^b^	0.01	0.043
Leu	0.19	0.16	0.20	0.15	0.02	0.197
Tyr	0.07	0.08	0.08	0.07	0.01	0.842
Phe	0.10	0.10	0.11	0.10	0.01	0.923
Lys	0.19 ^a,b^	0.18 ^a,b^	0.24 ^a^	0.12 ^b^	0.02	0.005
His	0.09	0.11	0.09	0.10	0.01	0.266
Arg	0.13	0.12	0.13	0.09	0.02	0.156
Pro	0.23	0.27	0.21	0.23	0.02	0.278

^1^ Control, a corn-soybean basal diet; TM, control diet +5% *Tenebrio molitor* powder; MDL, control diet +5% *Musca domestica* larvae powder; ZM, control diet +5% *Zophobas morio* powder. ^a,b^ Within a row, means sharing different superscript letters differ significantly (*p* < 0.05).

**Table 2 animals-10-01590-t002:** The effect of different insects protein on free amino acids profiles in jejunal mucosa of weanling pigs (µmol g^−1^).

Items	Treatments ^1^				SEM	*p*-Value
	Control	TM	MDL	ZM		
Day 28						
Asp	2.06	2.09	2.24	2.20	0.06	0.789
Thr	1.01	0.93	0.73	0.95	0.05	0.421
Ser	2.62	1.84	1.55	2.36	0.18	0.170
Glu	8.04	7.72	7.39	7.40	0.19	0.661
Gly	7.56	6.09	5.64	6.31	0.36	0.302
Ala	4.14	3.21	3.23	3.69	0.20	0.299
Cys	0.27 ^a^	0.18 ^b^	0.17 ^b^	0.19 ^b^	0.01	<0.001
Val	1.18 ^a^	0.96 ^b,c^	0.78 ^c^	1.15 ^a,b^	0.05	0.002
Met	0.58	0.46	0.44	0.59	0.03	0.222
Ile	0.66	0.50	0.46	0.66	0.04	0.113
Leu	1.59	1.28	1.15	1.63	0.09	0.169
Tyr	0.59 ^a^	0.47 ^a,b^	0.35 ^b^	0.57 ^a^	0.03	0.019
Phe	1.25	1.17	1.11	1.44	0.05	0.223
Lys	1.95 ^a^	1.50 ^b,c^	1.10 ^c^	1.72 ^a,b^	0.09	0.001
His	0.60	0.43	0.38	0.58	0.04	0.069
Arg	1.46	1.07	0.92	1.45	0.10	0.146
Pro	1.89	1.26	1.11	1.78	0.13	0.080
Day 56						
Asp	2.59 ^a^	1.90 ^b^	2.03 ^b^	2.31 ^a,b^	0.09	0.038
Thr	1.11 ^a^	0.81 ^b^	0.80 ^b^	0.91 ^a,b^	0.04	0.027
Ser	2.80 ^a^	1.75 ^c^	1.65 ^c^	2.40 ^b^	0.12	<0.001
Glu	8.07	7.09	7.06	7.97	0.18	0.051
Gly	6.98 ^a^	5.00 ^b^	4.85 ^b^	6.68 ^a^	0.26	<0.001
Ala	3.81 ^a^	2.89 ^b^	2.72 ^b^	3.84 ^a^	0.16	0.002
Cys	0.20	0.13	0.15	0.19	0.01	0.296
Val	1.37 ^a^	0.88 ^c^	0.88 ^c^	1.16 ^b^	0.05	<0.001
Met	0.68 ^a^	0.51 ^b,c^	0.45 ^c^	0.58 ^b^	0.02	<0.001
Ile	0.74 ^a^	0.52 ^c^	0.52 ^c^	0.64 ^b^	0.03	<0.001
Leu	1.86 ^a^	1.33 ^c^	1.28 ^c^	1.62 ^b^	0.07	<0.001
Tyr	0.75 ^a^	0.44 ^c^	0.38 ^c^	0.56 ^b^	0.04	<0.001
Phe	1.52 ^a^	1.42 ^a,b^	1.25 ^b^	1.49 ^a^	0.04	0.026
Lys	2.15 ^a^	1.41 ^c^	1.37 ^c^	1.75 ^b^	0.08	<0.001
His	0.63 ^a^	0.43 ^b^	0.39 ^b^	0.56 ^a^	0.03	<0.001
Arg	1.70 ^a^	1.10 ^c^	1.04 ^c^	1.44 ^b^	0.07	<0.001
Pro	2.05 ^a^	1.28 ^b^	1.14 ^b^	1.82 ^a^	0.10	<0.001

^1^ Control, a corn-soybean basal diet; TM, control diet +5% *Tenebrio molitor* powder; MDL, control diet +5% *Musca domestica* larvae powder; ZM, control diet +5% *Zophobas morio* powder. ^a,b,c^ Within a row, means sharing different superscript letters differ significantly (*p* < 0.05).

**Table 3 animals-10-01590-t003:** The effect of different protein from insects on free amino acids profiles in ileal mucosa of weanling pigs (µmol g^−1^).

Items	Treatment ^1^				SEM	*p*-Value
	Control	TM	MDL	ZM		
Day 28						
Asp	2.70	2.49	2.74	3.05	0.12	0.448
Thr	1.21 ^b^	1.29 ^b^	1.93 ^b^	2.91 ^a^	0.23	0.017
Ser	2.21 ^b,c^	2.00 ^c^	2.72 ^a,b^	2.96 ^a^	0.14	0.009
Glu	11.38	10.99	12.04	11.57	0.29	0.704
Gly	9.31	8.77	8.92	8.78	0.37	0.961
Ala	3.56	3.22	4.32	4.28	0.19	0.088
Val	1.11 ^b^	1.07 ^b^	1.40 ^a^	1.57 ^a^	0.07	0.005
Met	0.50 ^b^	0.51 ^b^	0.68 ^a^	0.80 ^a^	0.04	0.002
Ile	0.58 ^b^	0.54 ^b^	0.73 ^a^	0.84 ^a^	0.04	0.002
Leu	1.50 ^c^	1.40 ^c^	1.86 ^b^	2.21 ^a^	0.10	0.001
Tyr	0.62 ^c^	0.60 ^c^	0.77 ^b^	0.89 ^a^	0.04	0.002
Phe	1.07 ^b^	0.97 ^b^	1.19 ^b^	1.52 ^a^	0.07	0.005
Lys	1.58 ^b,c^	1.34 ^c^	1.81 ^b^	2.20 ^a^	0.11	0.002
His	0.53 ^b,c^	0.49 ^c^	0.63 ^a,b^	0.67 ^a^	0.03	0.020
Arg	1.30 ^b,c^	1.15 ^c^	1.55 ^a,b^	1.81 ^a^	0.09	0.009
Pro	1.41 ^a,b^	1.10 ^b^	1.59 ^a^	1.67 ^a^	0.09	0.043
Day 56						
Asp	2.48 ^b,c^	2.24 ^c^	2. 80 ^a,b^	2.90 ^a^	0.08	0.003
Thr	1.82	1.81	2.38	2.57	0.15	0.186
Ser	2.25 ^c^	2.52 ^c^	3.07 ^b^	3.70 ^a^	0.15	<0.001
Glu	10.79	10.97	11.06	11.69	0.16	0.227
Gly	8.16	9.15	8.81	9.72	0.22	0.062
Ala	3.23 ^b^	3.75 ^b^	3.88 ^b^	4.97 ^a^	0.19	0.003
Val	1.29 ^c^	1.33 ^b,c^	1.59 ^a,b^	1.86 ^a^	0.07	0.001
Met	0.57 ^c^	0.60 ^c^	0.72 ^b^	0.84 ^a^	0.03	<0.001
Ile	0.67 ^b^	0.68 ^b^	0.82 ^a,b^	0.95 ^a^	0.04	0.004
Leu	1.60 ^b^	1.66 ^b^	1.98 ^b^	2.40 ^a^	0.10	0.003
Tyr	0.68 ^c^	0.72 ^b,c^	0.86 ^a,b^	1.01 ^a^	0.04	0.001
Phe	1.04 ^b^	1.16 ^b^	1.21 ^b^	1.50 ^a^	0.06	0.011
Lys	1.70 ^c^	1.80 ^c^	2.16 ^b^	2.52 ^a^	0.09	<0.001
His	0.54 ^b^	0.57 ^b^	0.68 ^a^	0.73 ^a^	0.02	<0.001
Arg	1.39 ^c^	1.48 ^c^	1.77 ^b^	2.07 ^a^	0.08	<0.001
Pro	1.32 ^b^	1.35 ^b^	1.97 ^a^	1.79 ^a^	0.08	0.001

^1^ Control, a corn-soybean basal diet; TM, control diet +5% *Tenebrio molitor* powder; MDL, control diet +5% *Musca domestica* larvae powder; ZM, control diet +5% *Zophobas morio* powder. ^a,b,c^ Within a row, means sharing different superscript letters differ significantly (*p* < 0.05).

**Table 4 animals-10-01590-t004:** The effect of different protein from insects on free amino acids profiles in colonic mucosa of weanling pigs (µmol g^−1^).

Items	Treatments ^1^				SEM	*p*-Value
	Control	TM	MDL	ZM		
Day 28						
Asp	1.08 ^b^	1.50 ^a,b^	1.98 ^a^	1.72 ^a^	0.11	0.016
Thr	0.65	1.04	1.86	1.13	0.16	0.096
Ser	0.98 ^b^	1.68 ^a^	2.06 ^a^	1.76 ^a^	0.13	0.004
Glu	5.07 ^b^	6.17 ^a,b^	7.59 ^a^	6.21 ^a,b^	0.30	0.030
Gly	3.54	4.83	6.31	5.00	0.39	0.142
Ala	1.75 ^c^	2.44 ^b,c^	3.29 ^a^	2.56 ^a,b^	0.17	0.012
Val	0.66 ^c^	0.99 ^b^	1.30 ^a^	0.90 ^b,c^	0.07	0.004
Met	0.27 ^c^	0.44 ^b^	0.57 ^a^	0.51 ^a,b^	0.03	0.001
Ile	0.33 ^c^	0.50 ^b^	0.70 ^a^	0.56 ^b^	0.04	0.002
Leu	0.87 ^c^	1.31 ^b^	1.88 ^a^	1.47 ^b^	0.11	0.002
Tyr	0.28 ^c^	0.49 ^a,b^	0.61 ^a^	0.42 ^b^	0.04	0.004
Phe	0.68 ^c^	1.01 ^b^	1.46 ^a^	1.25 ^a,b^	0.09	0.002
Lys	0.60 ^c^	1.06 ^b^	1.34 ^a,b^	1.48 ^a^	0.10	0.001
His	0.27 ^c^	0.37 ^b^	0.40 ^b^	0.47 ^a^	0.02	0.001
Arg	0.48 ^b^	0.87 ^a^	1.14 ^a^	1.04 ^a^	0.08	0.003
Pro	0.50 ^c^	0.76 ^b^	0.91 ^b^	1.42 ^a^	0.09	<0.001
Day 56						
Asp	0.90	0.91	1.06	0.89	0.04	0.263
Thr	0.86	0.93	0.65	0.66	0.07	0.344
Ser	0.92 ^b^	1.27 ^a^	1.01 ^b^	1.16 ^a,b^	0.05	0.024
Glu	4.58	4.52	4.83	4.29	0.13	0.577
Gly	2.54	3.13	2.68	2.64	0.11	0.248
Ala	1.60	1.90	1.66	1.74	0.06	0.395
Val	0.65	0.76	0.61	0.61	0.02	0.078
Met	0.25 ^b^	0.33 ^a^	0.37 ^a^	0.37 ^a^	0.02	0.004
Ile	0.32	0.39	0.41	0.40	0.02	0.052
Leu	0.89	1.10	1.11	1.07	0.04	0.108
Tyr	0.27 ^b^	0.39 ^a^	0.25 ^b^	0.26 ^b^	0.02	0.002
Phe	0.72 ^c^	0.88 ^b^	1.15 ^a^	1.05 ^a^	0.05	<0.001
Lys	0.63 ^b^	0.93 ^a^	0.88 ^a^	0.92 ^a^	0.04	0.009
His	0.26	0.32	0.30	0.28	0.01	0.146
Arg	0.48 ^b^	0.69 ^a^	0.63 ^a,b^	0.68 ^a^	0.03	0.029
Pro	0.51 ^b^	0.68 ^a,b^	0.80 ^a^	0.88 ^a^	0.05	0.007

^1^ Control, a corn-soybean basal diet; TM, control diet +5% *Tenebrio molitor* powder; MDL, control diet +5% *Musca domestica* larvae powder; ZM, control diet +5% *Zophobas morio* powder. ^a,b,c^ Within a row, means sharing different superscript letters differ significantly (*p* < 0.05).

**Table 5 animals-10-01590-t005:** mRNA expression level of the amino acid transporter and sensing gene in the jejunal mucosa.

Items ^2^	Treatments ^1^				SEM	*p*-Value
	Control	TM	MDL	ZM		
Day 28						
TRPV1	1.00	0.73	0.87	0.80	0.08	0.703
FFAR3	1.00	1.20	0.54	0.76	0.16	0.591
TRPM5	1.00	2.45	3.07	5.70	0.83	0.395
mGluR	1.00	1.64	0.30	1.99	0.24	0.102
T1R1	1.00	1.78	1.16	1.22	0.22	0.559
T1R3	1.00	2.91	3.00	1.21	0.51	0.394
PLCβ2	1.00	1.10	0.73	1.00	0.15	0.908
GPRC6A	1.00 ^b^	0.95 ^b^	0.79 ^b^	2.71 ^a^	0.25	0.032
PAT1	1.00	1.71	1.06	1.97	0.20	0.320
PAT2	1.00	1.14	1.64	2.27	0.26	0.426
y+LAT1	1.00	0.77	0.90	0.88	0.03	0.195
SNAT2	1.00	1.08	2.38	0.45	0.31	0.320
GCN2	1.00	1.14	1.09	1.22	0.11	0.954
MAP4K3	1.00 ^a^	0.34 ^b^	1.15 ^a^	0.83 ^a,b^	0.12	0.043
PIK3C3	1.00	0.85	1.52	1.21	0.13	0.364
S6K1	1.00	1.05	1.16	1.79	0.13	0.195
Day 56						
TRPV1	1.00	0.91	0.51	0.50	0.12	0.333
FFAR3	1.00	0.87	0.60	0.58	0.11	0.464
TRPM5	1.00	2.88	1.40	0.80	0.45	0.376
mGluR	1.00 ^a,b^	1.49 ^a^	0.52 ^b^	0.27 ^b^	0.18	0.049
T1R1	1.00	1.41	1.05	0.74	0.12	0.263
T1R3	1.00	0.39	0.36	0.44	0.11	0.092
PLCβ2	1.00 ^a^	0.61 ^a,b^	0.65 ^a,b^	0.33 ^c^	0.07	0.001
GPRC6A	1.00 ^a^	1.09 ^a^	0.40 ^b^	0.35 ^b^	0.11	0.015
PAT1	1.00	0.90	0.75	0.87	0.09	0.842
PAT2	1.00 ^a,b^	1.63 ^a^	0.39 ^b^	0.36 ^b^	0.17	0.013
y+LAT1	1.00	0.85	0.78	1.09	0.11	0.784
SNAT2	1.00	1.62	2.20	1.26	0.22	0.257
GCN2	1.00	2.07	1.19	0.62	0.21	0.090
MAP4K3	1.00	1.60	0.45	0.71	0.16	0.058
PIK3C3	1.00	2.19	0.72	0.79	0.24	0.104
S6K1	1.00 ^b^	2.03 ^a^	1.22 ^b^	0.59 ^b^	0.16	0.006

^1^ Control, a corn-soybean basal diet; TM, control diet +5% *Tenebrio molitor* powder; MDL, control diet +5% *Musca domestica* larvae powder; ZM, control diet +5% *Zophobas morio* powder. ^2^ TRPV1, The transient receptor potential cation channel subfamily V member 1; FFAR3, free fatty acid receptor 3; TRPM5, transient receptor potential cation channel subfamily M member 5; mGluR, metabotropic glutamate receptors; T1R1, taste 1 receptor member 1; PLCβ2, Gβγ-mediated phospholipase C β2; GPRC6A, G protein-coupled receptor family C group 6 member A; PAT1, proton-coupled amino acid transporter 1; PAT2, proton-coupled amino acid transporter 2; Y+LAT1, L-type amino acid transporter 1; SNAT2, sodium dependent neutral amino acid transporter 2; GCN2, general control nonderepressible 2; MAP4K3, mitogen-activated protein 4 kinase 3; PIK3C3, phosphatidylinositol 3-kinase catalytic subunit type 3; S6K1, ribosome protein subunit 6 kinase 1. ^a,b,c^ Within a row, means sharing different superscript letters differ significantly (*p* < 0.05).

**Table 6 animals-10-01590-t006:** The mRNA expression level of the amino acid transporter and sensing gene in the ileal mucosa.

Items ^2^	Treatments ^1^				SEM	*p*-Value
	Control	TM	MDL	ZM		
Day 28						
TRPV1	1.00	0.74	0.74	0.93	0.10	0.774
FFAR3	1.00	2.30	1.46	2.49	0.29	0.244
TRPM5	1.00	0.74	0.75	1.45	0.12	0.167
mGluR	1.00	1.44	1.11	1.66	0.14	0.452
T1R3	1.00	0.72	1.12	1.10	0.11	0.581
PLCβ2	1.00	0.78	0.83	0.61	0.08	0.505
GPRC6A	1.00	1.33	0.99	1.14	0.09	0.547
PAT1	1.00	3.27	3.47	1.94	0.42	0.102
PAT2	1.00	2.31	3.10	0.66	0.35	0.067
Y+LAT1	1.00	1.55	2.71	2.43	0.32	0.270
SNAT2	1.00	1.66	1.22	1.02	0.13	0.146
GCN2	1.00	0.78	0.70	0.57	0.11	0.705
MAP4K3	1.00	0.93	1.01	0.98	0.12	0.997
PIK3C3	1.00	0.64	0.78	0.77	0.08	0.418
S6K1	1.00	0.63	1.24	0.75	0.12	0.365
Day 56						
TRPV1	1.00 ^b^	1.50 ^b^	3.40 ^a^	1.24 ^b^	0.34	0.023
FFAR3	1.00	0.92	1.60	1.87	0.17	0.126
TRPM5	1.00	1.13	0.72	1.12	0.15	0.777
mGluR	1.00	1.03	1.09	0.74	0.09	0.541
T1R3	1.00 ^b^	0.89 ^b^	2.26 ^a^	0.61 ^b^	0.23	0.027
PLCβ2	1.00 ^a,b^	0.55 ^b^	1.34 ^a^	0.50 ^b^	0.12	0.014
GPRC6A	1.00	0.69	1.08	0.96	0.13	0.803
PAT1	1.00	2.85	1.21	2.02	0.28	0.131
PAT2	1.00	2.43	1.15	0.81	0.34	0.299
Y+LAT1	1.00	0.99	1.86	1.05	0.23	0.473
SNAT2	1.00	0.88	0.61	0.84	0.11	0.728
GCN2	1.00	0.80	1.13	0.62	0.12	0.435
MAP4K3	1.00	1.29	1.06	0.92	0.10	0.610
PIK3C3	1.00	0.86	1.43	0.89	0.10	0.174
S6K1	1.00	0.63	0.48	0.41	0.10	0.235

^1^ Control, a corn-soybean basal diet; TM, control diet + 5% *Tenebrio molitor* powder; MDL, control diet + 5% *Musca domestica* larvae powder; ZM, control diet + 5% *Zophobas morio* powder. ^2^
*TRPV1*, The transient receptor potential cation channel subfamily V member 1; FFAR3, free fatty acid receptor 3; TRPM5, transient receptor potential cation channel subfamily M member 5; mGluR, metabotropic glutamate receptors; T1R1, taste 1 receptor member 1; PLCβ2, Gβγ-mediated phospholipase C β2; GPRC6A, G protein-coupled receptor family C group 6 member A; PAT1, proton-coupled amino acid transporter 1; PAT2, proton-coupled amino acid transporter 2; Y+LAT1, L-type amino acid transporter 1; SNAT2, sodium dependent neutral amino acid transporter 2; GCN2, general control nonderepressible 2; MAP4K3, mitogen-activated protein 4 kinase 3; PIK3C3, phosphatidylinositol 3-kinase catalytic subunit type 3; S6K1, ribosome protein subunit 6 kinase 1. ^a,b^ Within a row, means sharing different superscript letters differ significantly (*p* < 0.05).

**Table 7 animals-10-01590-t007:** The mRNA expression level of the amino acid transporter and sensing gene in the colonic mucosa.

Items ^2^	Treatments ^1^				SEM	*p*-Value
	Control	TM	MDL	ZM		
Day 28						
mGluR	1.00	2.60	2.31	2.21	0.48	0.759
PLCβ2	1.00	0.76	1.04	0.85	0.08	0.633
GPRC6A	1.00	4.29	0.42	4.79	1.08	0.479
PAT1	1.00	1.28	1.72	1.73	0.17	0.487
PAT2	1.00	1.49	1.59	2.35	0.28	0.549
SNAT2	1.00	1.61	0.48	2.89	0.39	0.322
GCN2	1.00	1.37	0.75	0.89	0.11	0.160
MAP4K3	1.00	1.54	1.15	1.56	0.25	0.854
PIK3C3	1.00	1.69	0.99	1.27	0.15	0.246
S6K1	1.00	1.06	0.74	1.12	0.07	0.362
Day 56						
mGluR	1.00	0.48	0.67	0.58	0.10	0.465
PLCβ2	1.00	0.51	0.56	0.54	0.08	0.056
GPRC6A	1.00 ^a^	0.29 ^b^	0.19^b^	0.46 ^a,b^	0.11	0.022
PAT1	1.00	0.87	0.71	0.93	0.06	0.349
PAT2	1.00	0.42	0.52	0.18	0.14	0.250
SNAT2	1.00 ^b^	0.70 ^b^	0.98 ^b^	2.07 ^a^	0.17	0.045
GCN2	1.00	0.92	0.66	0.60	0.10	0.490
MAP4K3	1.00	0.98	0.75	0.90	0.10	0.838
PIK3C3	1.00	1.21	1.28	1.13	0.11	0.835
S6K1	1.00	1.13	0.75	0.91	0.08	0.398

^1^ Control, a corn-soybean basal diet; TM, control diet + 5% *Tenebrio molitor* powder; MDL, control diet + 5% *Musca domestica* larvae powder; ZM, control diet + 5% *Zophobas morio* powder. ^2^ TRPV1, The transient receptor potential cation channel subfamily V member 1; FFAR3, free fatty acid receptor 3; TRPM5, transient receptor potential cation channel subfamily M member 5; mGluR, metabotropic glutamate receptors; T1R1, taste 1 receptor member 1; PLCβ2, Gβγ-mediated phospholipase C β2; GPRC6A, G protein-coupled receptor family C group 6 member A; PAT1, proton-coupled amino acid transporter 1; PAT2, proton-coupled amino acid transporter 2; Y+LAT1, L-type amino acid transporter 1; SNAT2, sodium dependent neutral amino acid transporter 2; GCN2, general control nonderepressible 2; MAP4K3, mitogen-activated protein 4 kinase 3; PIK3C3, phosphatidylinositol 3-kinase catalytic subunit type 3; S6K1, ribosome protein subunit 6 kinase 1. ^a,b^ Within a row, means sharing different superscript letters differ significantly (*p* < 0.05).

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
