# Peer review of "Dietary Insect Powder Protein Sources Improve Protein Utilization by Regulation on Intestinal Amino Acid-Chemosensing System"

_animals, 2020, doi:10.3390/ani10091590_

Round 1

Reviewer 1 Report

The paper undoubtedly contributes to spread the knowledge of alternative feeding components. The discussion is well written and conclusions are clear. Also, there is not much to criticise about the execution of the study. However, there are some minor flaws in introduction and method description. 

First of all Latin names throughout the text should be in italics and the capital letter should be used only in the taxon name - e.g. Tenebrio molitor (L18, 58). I also recommend to highlight that authors (as far as I get it) used worms/larvae of the experimental insects. Therefore, the sentence at L18 should read: "..including larvae of, Musca domestica (MDL),  Tenebrio molitor (TM) and Zophobas morio (ZM)." 

L50: I dont think that the protein content of insects generally ranges between 40-75 g/kg DM. It might be cca 20-80, but generally it is somewhere between 40-60. Maybe, the information on the protein content from some recently published review on the nutritional values of insects might be used to support this statement.  Also, some recent sources should be used to describe factors influencing the nutritional values/protein content such as species, diet, developmental stage etc.

L58: "belong among the most common"

L70: apparent ileal digestability (AID) - and generally, the abbreviations should be explained when first noted throughout the text.

L79: insect powder

L80-90: I dislike this paragraph, it seems to be confusing and requires further attention. Why do authors mention nutritional value of insects here? It seems to be the result. And I also do not understand, why do authors not the information about the insect origin twice in the same part of methods. Were they provided by the whole insects or powder? Or both?

L188: The table should not be divided on two pages.

Author Response

Response to Reviewer 1

Comments

  1. First of all Latin names throughout the text should be in italics and the capital letter should be used only in the taxon name - e.g.Tenebrio molitor(L18, 58). I also recommend to highlight that authors (as far as I get it) used worms/larvae of the experimental insects. Therefore, the sentence at L18 should read: "..including larvae of, Musca domestica (MDL), Tenebrio molitor(TM) and Zophobas morio(ZM)."

Response: Line 18-19, 30-31 and 63-64: “Tenebrio Molitor (TM), Musca Domestica Larvae (MDL) and Zophobas Morio (ZM)was revised as “Tenebrio molitor (TM), Musca domestica larvae (MDL) and Zophobas morio (ZM)”. Moreover, it has been revised in the supplemental Table1 and Table 2.

  1. L50: I dont think that the protein content of insects generally ranges between 40-75 g/kg DM. It might be cca 20-80, but generally it is somewhere between 40-60. Maybe, the information on the protein content from some recently published review on the nutritional values of insects might be used to support this statement. Also, some recent sources should be used to describe factors influencing the nutritional values/protein content such as species, diet, developmental stage etc.

Response: “Insects are being considered as a renewable source of protein. In general, the insect contains protein from 40 to 75 g/100 g dry weight” was improved as “Insects can turn low-grade bio-waste into high quality proteins indicating that insects as a renewable protein rich feed ingredient is technically feasible.” Besides, the protein content of insects generally ranges between 40-75 g/kg DM was reported in previous studies. I believe this is the largest ranges of insect protein levels.

  1. L58: "belong among the most common"

Response: Line 61-62“Tenebrio Molitor (TM), Musca Domestica Larvae (MDL) and Zophobas Morio (ZM) are the most common species of insects as feedstuff in the animal food market”has been revised as “Tenebrio molitor (TM), Musca domestica larvae (MDL) and Zophobas morio (ZM) belong among most common species of insects as feedstuff in the animal food market”.

  1. L70: apparent ileal digestability (AID) - and generally, the abbreviations should be explained when first noted throughout the text.

Response: L74-75 “In the previous study, we found that plasma concentrations of total protein and albumin were reduced by dietary ZM whereas Met AID was improved in the pigs” has been revised as “In the previous study, we found that plasma concentrations of total protein and albumin were reduced by dietary ZM whereas methionine (Met) apparent ileal digestability (AID) was improved in the pigs”.

  1. L79: insect powder

Response: L84 “2.1. Preparation of iInsect powder” has been revised as “2.1. Preparation of insect powder”. And it was removed.

  1. L80-90: I dislike this paragraph, it seems to be confusing and requires further attention. Why do authors mention nutritional value of insects here? It seems to be the result. And I also do not understand, why do authors not the information about the insect origin twice in the same part of methods. Were they provided by the whole insects or powder? Or both?

Response: These insects were provided by powder, which has been added in Line 102-103. “TM, MDL and ZM” was revised as “TM powder, MDL powder and ZM powder”. And the nutritional value of insects can't be used as a result, because it is a reference to previous studies. And this paragraph was removed.

  1. L188: The table should not be divided on two pages

Response: All tables have been arranged in one pages (Table 4, 5, 6 and 7). I will pay attention to this problem.

Reviewer 2 Report

The authors investigated the effects of dietary inclusion of three different insect meal powder (TM, MDL and ZM) on free amino acids profile (in plasma and intestinal mucosa) and amino acid transporter and sensing gene expression in intestinal mucosa (jejunal, ileal and colonic mucosa).

The main conclusion of the present manuscript is that insect powder improved amino acid transportation in the intestine through regulating the sensing gene and mTOR signal pathway in intestinal mucosa by targeting different receptors. The problem in this paper is that there are not enough results and discussion to support this conclusion; no correlation was done in this study related to the growth, nutrient utilization (protein and AA) and digestibility of protein and AA reported in the study of Ji et al., 2016. Also, the hypothesis of the present study is based on a previous result (Ji et al., 2016), where it reported that dietary ZM reduced total protein and albumin in the plasma and improved met AID (68-70). However, Met AID is improved when feeding piglets dietary TM and MDL. Further, to conclude that insect meal powder improved AA transportation in the intestine through mTOR signaling pathway, an additional test by using the protein expression of mTOR might give more information.             

-Simple summary need to be revised –>” The mechanical effect of insect protein on amino acid metabolism in pigs was need to be explored.”

-L 52 “Previous studies showed that most insects had a good composition as human food 4, 5” is it edible insect? Nutrient composition? Does the references 4 and 5 show similar nutrient values for edible insects and human food?

-L56 However, the molecular  effect of dietary.

-L70: definition for AID before abbreviation

-L73: correct “ we hypothesis

-L79:Remove extra I for insect

-L81: obtained in oven ?

-L84-88: why the results are given in the materials and methods section?

-L 79-104: preparation of insect powder and animal housing and treatment are described in the study of Ji et al., 2016. No need to repeat but just mentioning that the feed production and feeding trials are reported in more detail elsewhere (9).

L 170: ..whereas Thr, Glu and gly

L 180-182: “ As the result in ……..with insect supplementation” and L 182-184: “Similar to the results….compared to the control diets” it’s the same results repeated twice.

-The primers sequence and accession numbers are not reported.

L246” The protein from the insect is deemed to have favourable conversion efficacy 18.”the insect have a favorable feed conversion efficiency because they are cold blooded, how the proteins from the insect have favorable conversion efficiency?

A big part of the discussion is based on an explanation of the role of the genes (from L 258-284 and L293-300), however there are not so much speculations of the obtained results in relation to other studies. Also most of the differences in AA and gene expression are observed in the second phase compared to the first phase and her also there are not a discussion about it. Discussion is missing also on the differences between the three insect meal, on the AA composition and on the molecular basis of feeding piglets with three different sources of proteins.

Author Response

Response to Reviewer 2

Comments

  1. Simple summary need to be revised –>” The mechanical effect of insect protein on amino acid metabolism in pigs was need to be explored.”

Response: Line 20-21 “The mechanical effect of insect protein on amino acid metabolism in pigs was need to be explored.” was revised as “The molecular effect of insect protein on amino acid metabolism in pigs needs to be explored.”

  1. L 52 “Previous studies showed that most insects had a good composition as human food 4, 5” is it edible insect? Nutrient composition? Does the references 4 and 5 show similar nutrient values for edible insects and human food?

Response: Line 55-56 “Previous studies showed that most insects had a good composition as human food 4,5.” was was revised as “Previous studies showed that most insects had high nutritional qualities as human food 4,5.” The references 4,5 show that most insects had a good nutritient composition, and constituted a significant component of diet among the people of south western Nigeria.

  1. L56 However, the molecular effect of dietary.

Response: It is a spelling error. L58-59 “However, the mechanical effect of dietary insects on amino acid transportation is not well known.” has been revised as “However, the molecular effect of dietary insects on amino acid transportation is not well known”.

  1. L70: definition for AID before abbreviation

Response: L74-75 “In the previous study, we found that plasma concentrations of total protein and albumin were reduced by dietary ZM whereas Met AID was improved in the pigs” has been revised as “In the previous study, we found that plasma concentrations of total protein and albumin were reduced by dietary ZM whereas methionine (Met) apparent ileal digestability (AID) was improved in the pigs”.

  1. L73: correct “ we hypothesis

Response: Line 78 “hypothesis” was revised as “hypothesize”.

  1. L79:Remove extra I for insect

Response: Line 84“2.1. Preparation of iInsect powder” has been revised as “2.1. Preparation of insect powder”. But it was removed.

  1. L81: obtained in oven ?

Response: Line 84 “For enzyme destruction, mature larvae were obtained in oven at 105℃ for 15 min, plus at 65℃ for 24 h.” was revised as “Mature larvae baked in oven at 105℃ for 15 min, plus at 65℃ for 24 h to destroy enzymes.” It was removed.

  1. L84-88: why the results are given in the materials and methods section?

Response: The nutritional value of insects can't be used as a result. It is a reference to previous studies. It was removed.

  1. L79-104: preparation of insect powder and animal housing and treatment are described in the study of Ji et al., 2016. No need to repeat but just mentioning that the feed production and feeding trials are reported in more detail elsewhere (9).

Response: “2.1 Preparation of insect powder”and “2.2. Animal housing and treatment” were rewritten.

  1. L170: ..whereas Thr, Glu and gly

Response: Line 181 “whereas Thr Glu and were the only AAs that were unaffected by dietary insect powder in 2nd phase.” was revised as “whereas Thr, Glu and Gly were the only AAs that were unaffected by dietary insect powder in 2nd phase.”

  1. L180-182: “ As the result in ……..with insect supplementation” and L 182-184: “Similar to the results….compared to the control diets” it’s the same results repeated twice.

Response: L193-194: “Similar to the results….compared to the control diets” was removed.

  1. The primers sequence and accession numbers are not reported.

Response: The primers sequence and accession numbers were added and showed in Table S3 (Supplemental materia, Line 463).

  1. L246” The protein from the insect is deemed to have favourable conversion efficacy 18.”the insect have a favorable feed conversion efficiency because they are cold blooded, how the proteins from the insect have favorable conversion efficiency?

Response: L276 “The protein from the insect is deemed to have favourable conversion efficacy” was revised as “The protein from the insect is in high in essential amino acids, which deemed to have favourable conversion efficacy”. Moreover. previous study found that Each insect and life stage showed EAAI values higher than 1, and apparent faecal digestibility of crude protein was similar with the soybean meal diet (reference 18).

  1. A big part of the discussion is based on an explanation of the role of the genes (from L 258-284 and L293-300), however there are not so much speculations of the obtained results in relation to other studies. Also most of the differences in AA and gene expression are observed in the second phase compared to the first phase and her also there are not a discussion about it. Discussion is missing also on the differences between the three insect meal, on the AA composition and on the molecular basis of feeding piglets with three different sources of proteins.

Response: There are few studies on the molecular mechanism of dietary insects powders on amino acid metabolism in piglets. And I don not think this disscussion about the differences between the three insect meal, on the AA composition and on the molecular basis of feeding piglets was a priority. The discussion mainly focuses on which insect protein significantly affected the amino acid metabolism signal pathway in piglets.

Reviewer 3 Report

Comments to the Authors of manuscript number: animals-891436 entitled “Dietary Insect Powder Protein Sources Improve Protein Utilization by Regulation on Intestinal Amino Acid-chemosensing System”. 1. Authors focused on the replacement of common form of protein with insects protein. They wrote L 104 that some data were presented early – reference 9. And, here we have a problem, Reference 9 concerns to the paper based on the same study with the same experimental protocol and piglets. In my mind, Authors should add this information in present manuscript. It is not a separate study, but based on earlier one material was collected and They were waiting for few years. 2. Moreover, NCR recommendation in both cases should be from 2012y. This recommendation is the earliest, and previous study was conducted in 2016y. Why Author used old recommendation? It should be explained. 3. L 26 L93 – the study protocol should be described clear, now it is problem with understanding how many piglets were involved in the study. 4. L 56- what is “the mechanical effect”. What is the meaning? It should be described. 5. L 70 – Authors used the abbreviation of “AID”, but it is not explained. Of course, they explain it in previous paper in the introduction, but it is a new one. 6. L 88 “Experimenta insects” ? 7. L 88 -89 the repetition of L 80 8. the part of “2.2. Animal housing and treatment” should be rewritten because now it is a copy from the reference 9. 9. Why Authors given 5% of powdered insects? It was based on preliminary study? Why not more or less? It should be described. 10. L 104 Why Authors wrote that “BW, ADG, and F:G ratio as described previously (data not shown in the present study)” It means that these data are not presented. However Authors refer to not published data. Why? It should be added or corrected.

Author Response

Response to Reviewer 3

Comments

  1. Authors focused on the replacement of common form of protein with insects protein. They wrote L 104 that some data were presented early – reference 9. And, here we have a problem, Reference 9 concerns to the paper based on the same study with the same experimental protocol and piglets. In my mind, Authors should add this information in present manuscript. It is not a separate study, but based on earlier one material was collected and They were waiting for few years.

Response: It has been added in present manuscript (Line 98). And the “Materials and methods” was revised.

  1. Moreover, NCR recommendation in both cases should be from 2012y. This recommendation is the earliest, and previous study was conducted in 2016y. Why Author used old recommendation? It should be explained.

Response: This experimental feed was configured earlier 2012y.

  1. L 26 L93 – the study protocol should be described clear, now it is problem with understanding how many piglets were involved in the study.

Response: Line 27 “Thirty-six weanling piglets were assigned to 1 of 4 diets for two phases (Days 1-28 and Days 29-56),” was revised as “A total of 144 weanling piglets were randomly assigned tofour experimental diets for two phases (Days 1-28 and Days 29-56),”.

Line 99-100 “A total of 144 Duroc × Landrace × Yorkshire piglets with an average body weight of 4.74 kg were weaned at 14 ± 2 days of age, and then randomly assigned to one of 4 groups based on BW and sex.” was revised as “one hundred forty-four piglets (Duroc × Landrace × Yorkshire, 4.74 ± 0.01 kg body weight) were randomly assigned to four groups.”

  1. L 56- what is “the mechanical effect”. What is the meaning? It should be described.

Response: It is a spelling error. L59-60 “However, the mechanical effect of dietary insects on amino acid transportation is not well known.” has been revised as “However, the molecular effect of dietary insects on amino acid transportation is not well known”.

  1. L 70 – Authors used the abbreviation of “AID”, but it is not explained. Of course, they explain it in previous paper in the introduction, but it is a new one.

Response: L74-75 “In the previous study, we found that plasma concentrations of total protein and albumin were reduced by dietary ZM whereas Met AID was improved in the pigs” has been revised as “In the previous study, we found that plasma concentrations of total protein and albumin were reduced by dietary ZM whereas methionine (Met) apparent ileal digestability (AID) was improved in the pigs”.

  1. L 88 “Experimenta insects” ? 7. L 88 -89 the repetition of L 80

Response:Experimenta insects were provided by Guangdong Entomological Institute (China).” was revised as “Experimental insects were provided by Guangdong Entomological Institute (China).” But the description of this sentence is repeated, so delete.

  1. the part of “2.2. Animal housing and treatment” should be rewritten because now it is a copy from the reference

Response:2.1 Preparation of insect powder”and “2.2. Animal housing and treatment” were rewritten.

  1. Why Authors given 5% of powdered insects? It was based on preliminary study? Why not more or less? It should be described.

Response: According to the NRC(1998) requirements for weaning piglets (4.72kg BW), the The addition of powdered insects meal should be consistent with plasma protein powder in the diet for comparing the effects.

  1. L 104 Why Authors wrote that “BW, ADG, and F:G ratio as described previously (data not shown in the present study)” It means that these data are not presented. However Authors refer to not published data. Why? It should be added or corrected.

Response: “There were no significant differences between the experimental groups supplemented with dried insect powder and the control group concerning BW, ADG, and F:G ratio as described previously (data not shown in the present study) 9. ” was removed. We don't think it needs to be presented.

Round 2

Reviewer 3 Report

Dear Authors,

The manuscript is corrected and can be published in this form. However, I have still one question.

If etical license "2016-8A" is probably from 2016y, how this study was performed earlier before NRC from 2012?